# RIViT-seq enables systematic identification of regulons of transcriptional machineries

Hiroshi Otani [1,2✉] & Nigel J. Mouncey [1,2✉]

Transcriptional regulation is a critical process to ensure expression of genes necessary for growth and survival in diverse environments. Transcription is mediated by multiple transcription factors including activators, repressors and sigma factors. Accurate computational prediction of the regulon of target genes for transcription factors is difficult and experimental identification is laborious and not scalable. Here, we demonstrate regulon identification by in vitro transcription-sequencing (RIViT-seq) that enables systematic identification of regulons of transcription factors by combining an in vitro transcription assay and RNA-sequencing. Using this technology, target genes of 11 sigma factors were identified in *Streptomyces coelicolor* A3(2). The RIViT-seq data expands the transcriptional regulatory network in this bacterium, discovering regulatory cascades and crosstalk between sigma factors. Implementation of RIViT-seq with other transcription factors and in other organisms will improve our understanding of transcriptional regulatory networks across biology.

[1] US Department of Energy Joint Genome Institute, Lawrence Berkeley National Laboratory, Berkeley, CA 94720, USA. [2] Environmental Genomics and Systems Biology Division, Lawrence Berkeley National Laboratory, Berkeley, CA 94720, USA. ✉email: hotani@lbl.gov; nmouncey@lbl.gov

Transcription is one of the most fundamental steps of gene expression in all living organisms that ensures genes required under given environmental conditions are expressed. Therefore, transcriptional regulation is a crucial system for proliferation, sensing and adapting to the environment, and communicating and cooperating with the surrounding organisms and cells, and any aberrant regulation may lead to undesired consequences, such as disease or cell death. A gene is transcribed to an RNA molecule, or a transcript, by an RNA polymerase enzyme complex. In eukaryotes, three distinct RNA polymerases transcribe different types of genes. In bacteria, however, only one type of RNA polymerase is responsible for transcribing every gene. Bacterial RNA polymerase consists of a core enzyme complex and a sigma factor. A core enzyme is comprised of five subunits and is responsible for RNA synthesis during transcription without preference for DNA sequence. During the transcription initiation process, the dissociable subunit of RNA polymerase, sigma factor, directs RNA polymerase to specific promoters and initiates transcription, conferring promoter selectivity to RNA polymerase[1]. Though all bacteria encode at least one principal sigma factor that is responsible for directing transcription of housekeeping genes, many bacteria encode multiple alternative sigma factors that exhibit varying promoter selectivity and specificity and initiate transcription of different sets of genes linked to specific functions such as stress response and cellular differentiation. In *Escherichia coli*, a total of 7 sigma factors are encoded in the genome and each of them directs the transcription of genes with specific functions. For example, $\sigma^{70}$ is the principal sigma factor and $\sigma^E$ controls the expression of genes involved in extracytoplasmic stress response[2]. While some model organisms including *E. coli* and *Bacillus subtilis* encode around 10 sigma factors, several organisms such as soil actinomycetes encode a far greater number of sigma factors. *Streptomyces coelicolor* A3(2), a soil actinomycete known to produce a wide variety of secondary metabolites, encodes 61 proteins that possess the minimum set of domains required for the sigma factor function that are classified into four subfamilies or groups of the $\sigma^{70}$ family (Supplementary Fig. 1) (Hiroshi Otani, Daniel W. Udwary and Nigel J. Mouncey, personal communications 2022). So far, only 26 sigma factors have been experimentally characterised for genes they regulate or the resulting biological function. Of them, at least one target gene has been identified for 12 sigma factors, including two sigma factors characterised only or primarily in related organisms, *Streptomyces griseus* and *Streptomyces venezuelae* (Supplementary Table 1)[3–8]. The activity of many sigma factors is controlled by cognate anti-sigma factors, proteolysis and other protein domains present in N- or C-terminal extension, and not all the sigma factors are active under laboratory growth conditions[9]. As such, characterisation of sigma factors through gene deletion, chromatin immunoprecipitation-sequencing (ChIP-seq) and trascriptomics does not guarantee the identification of their target genes, hindering their systematic characterisation.

In this study, we demonstrate a high throughput technology, regulon identification by in vitro transcription-sequencing (RIViT-seq), which enables systematic characterisation of transcriptional machineries for transcription factors of interest. In this assay, transcriptional machinery, or RNA polymerase, is reconstituted by combining its components such as a core enzyme and a sigma factor and an in vitro transcription assay is performed to transcribe the regulon of the transcription factor. These RNA molecules specifically produced by the reconstituted enzyme complex are identified by RNA-sequencing. We applied RIViT-seq to 13 purified sigma factors encoded in *S. coelicolor* A3(2), successfully identified at least one target gene for 11 sigma factors, and expanded the transcriptional regulatory network. Applying this technology to other proteins involved in transcriptional regulation

such as transcriptional regulators should simplify the identification of their regulons and facilitate expanding transcriptional regulatory networks.

## Results

**Development of RIViT-seq.** RIViT-seq consists of three steps: (i) in vitro transcription assay using a mixture of RNA polymerase core enzyme, purified sigma factor, genomic DNA and NTP mixture; (ii) whole transcriptomics by RNA sequencing; (iii) determination of 5′-ends by 5′-end sequencing (Fig. 1a). In the in vitro transcription assay step, an RNA polymerase holoenzyme complex was reconstituted by mixing *E. coli* RNA polymerase core enzyme and the sigma factor of interest, which recognises the sigma factor-specific promoter sequences. *E. coli* RNA polymerase core enzyme was previously used for in vitro transcription assays with sigma factors from diverse bacteria including streptomycetes[4,10–14]. Notably, one study demonstrated *E. coli* RNA polymerase core enzyme exhibiting similar activity to the mycobacterial RNA polymerase core enzyme at a mycobacterial promoter[11]. A mixture of genomic DNA digested by four different restriction enzymes was used as the template DNA of in vitro transcription. The NTP mixture was added to initiate the in vitro transcription reaction. Following the in vitro transcription reaction, the genomic DNA was digested by DNase, ERCC RNA Spike-in Mix was added as normalisation controls and the RNA molecules were purified. To optimise the in vitro transcription reaction conditions, two sigma factors, ShbA and SigR, which are known to initiate transcription at the *hrdB* and *trxB* promoters, respectively, were used[4,15]. Recombinant ShbA and SigR with a C-terminal hexahistidine tag were overproduced in *E. coli*, purified and used to determine the optimal concentrations of the RNA polymerase core enzyme, sigma factor and genomic DNA (Supplementary Fig. 2). Quantitative RT-PCR was used to measure abundances of the *hrdB*, *trxB* and ERCC transcripts, and the relative abundances of the *hrdB* and *trxB* transcripts were calculated using the ERCC transcripts as the normalisation controls. Because the RNA polymerase core enzyme is able to initiate transcription non-specifically, especially from ends of DNA fragments, the RNA polymerase core enzyme with no sigma factor was also used in order to determine the quantity of the transcripts produced non-specifically. The signal level increased by about 2.5 times for *hrdB* and *trxB* by the addition of ShbA and SigR, respectively, compared to the no sigma factor control (Fig. 1b) and the extent of signal increase was similar irrespective of the four ERCC transcripts that were used for normalisation (Fig. S3). These four ERCC transcripts represent a diverse range of transcript numbers (1–150 attomoles).

Subsequently, two separate types of Illumina sequencing libraries were created from these in vitro transcription samples, for whole transcriptomics and 5′-end sequencing (Fig. 1a). For whole transcriptomics, transcripts were fragmented and both 5′- and 3′-ends of transcripts were adaptor-ligated, followed by reverse transcription, amplification and sequencing. For 5′-end sequencing, transcripts were dephosphorylated and adaptors were ligated to only 5′-ends. The adaptor-ligated transcripts were reverse transcribed using random hexamers with the 3′ adaptor sequence, and the resulting cDNAs were amplified by PCR and sequenced. Similar to the quantitative RT-PCR data, the relative abundances of the *hrdB* and *trxB* transcripts increased by 2.2 and 2.7 times, respectively (Fig. 1c). In addition, a transcription start site (TSS) was determined within 2 nt of the previously identified TSS for each gene (Fig. 1d)[16,17].

We then searched for other genes directly transcribed by ShbA and SigR. Because the RNA polymerase core enzyme is capable of initiating transcription randomly, only genes satisfying both of

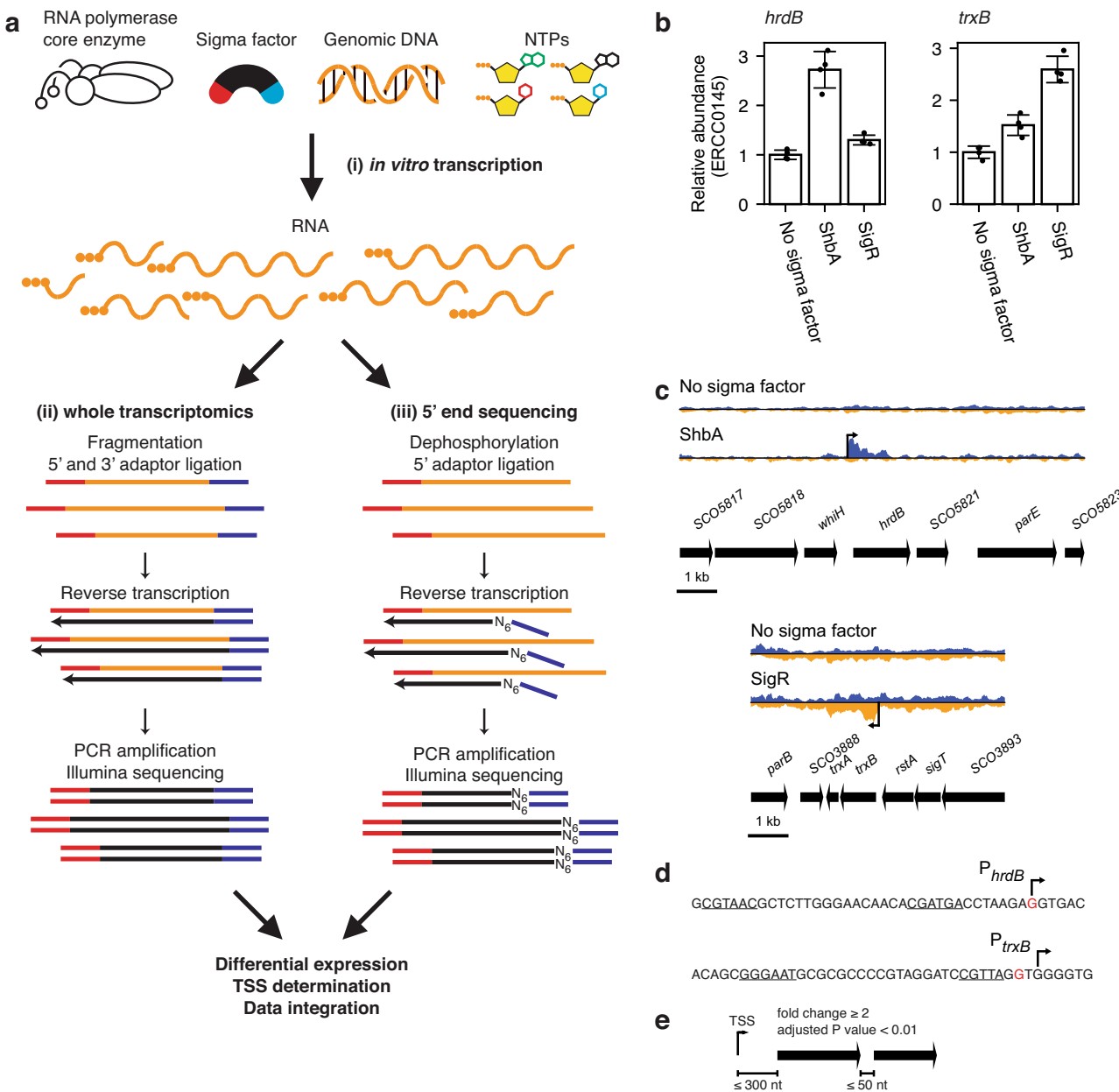

**Fig. 1 Development of RIViT-seq. a** Scheme of RIViT-seq. RIViT-seq consists of (i) in vitro transcription assay, (ii) whole transcriptomics and (iii) 5′-end sequencing. **b** Verification of the in vitro transcription reactions with ShbA and SigR by quantitative RT-PCR. The ERCC0145 transcript was used as the normalisation control. Values are relative abundances of the *hrdB* and *trxB* transcripts of the samples with ShbA or SigR compared to the mean abundance of the "No sigma factor" samples. Error bars are standard deviations ($n = 4$ independent experiments). **c** Transcriptional profile of the *hrdB* and *trxB* loci. In blue and orange are normalised read counts on the sense (left to right) and antisense (right to left) strands, respectively, of the in vitro transcripts detected by whole transcriptomics. Bent arrows are TSSs of the in vitro transcripts detected by 5′-end sequencing. **d** Promoter of *hrdB* recognised by ShbA (upper) and *trxB* recognised by SigR (bottom). Bent arrows are TSSs determined by 5′-end sequencing. Red letters indicate previously identified TSSs. Underlined letters are previously determined −35 and −10 regions. **e** Criteria used to determine target genes and operons. Fold changes and adjusted *P* values were determined by using the *DESeq2* package.

the following criteria were considered target genes of a sigma factor: (i) genes determined upregulated by whole transcriptomics and (ii) genes of which a proximal TSS is detected by 5′-end sequencing. For criterion (i), fold change of ≥2 and adjusted *P* value of <0.01 were used as the thresholds. TSSs were determined by 5′-end sequencing for criterion (ii) using the procedure described in Methods. By integrating these two datasets, a gene was determined a target of the sigma factor if it was upregulated and at least one TSS was identified within 300 nt upstream from its initiation codon. As multiple genes may be transcribed from a

single promoter in prokaryotes, an additional gene was considered a target if it was upregulated and located within 50 nucleotides downstream of a direct target gene oriented in the same direction (Fig. 1e). Using these criteria, *shbA* was determined to be the only other target gene of ShbA (Table 1; Supplementary Table 7). A total of 71 genes were determined as target genes of SigR including known target genes of SigR such as *hrdD*, and *moeB* (Table 1; Supplementary Table 9)[18,19]. In addition, direct transcriptional dependence of *rbpA*, encoding the RNA polymerase binding protein, on SigR was confirmed.

| Table 1 Number of target genes identified. | | | | |
|---|---|---|---|---|
| Locus tag | Symbol | Number of target genes identified in this study[a] | | Total Number of target genes[b] |
| | | Full-length | Truncated | |
| SCO0864 | | 3 | NA | 3 |
| SCO2742 | | 1 | 7 | 8 |
| SCO3068 | SigI | 40 | NA | 40 |
| SCO3626 | | 0 | NA | 0 |
| SCO4035 | SigF | 3 | NA | 4 |
| SCO4769 | ShbA | 2 | NA | 2 |
| SCO4895 | | 15 | NA | 15 |
| SCO5216 | SigR | 71 | NA | 134 |
| SCO5243 | SigH | 27 | 33 | 47 |
| SCO5820 | HrdB | 1 | 14 | 20 |
| SCO6239 | | 2 | NA | 2 |
| SCO6520 | SigK | 2 | NA | 2 |
| SCO7099 | | NA | 0 | 0 |

[a]NA: not applicable.
[b]The numbers include previously identified target genes.

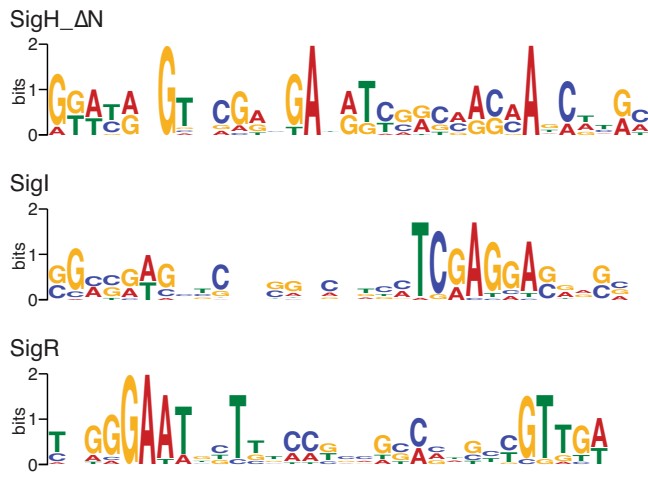

**Fig. 2 Sigma factor recognition motifs.** Sequence logos of the consensus sequences of the promoters recognised by SigH_ΔN, SigI and SigR. The logo is an illustration of the position weight matrix of the promoter sequence alignment.

Though direct involvement of SigR on the transcriptional initiation of *rbpA* was previously suggested by transcriptional analyses, it was not biochemically verified such as an in vitro transcription assay or chromatin immunoprecipitation assay[20]. A number of known target genes of SigR were not detected by RIViT-seq, similar to the previous studies which also conducted in vitro transcription assays[21,22]. This could be because their signal was obscured by the high level of the background transcription generated by the RNA polymerase core enzyme as the RNA polymerase core enzyme is known to initiate transcription from the ends of DNA fragments. For example, *trxA* was not determined a target gene because its fold change was 1.8, below the cut-off threshold that was applied. Another possibility is the lack of additional factor(s) in the in vitro transcription assay which are required to initiate the transcription of SigR target genes. WblC, a WhiB-family transcriptional activator, promotes transcription from one of the SigR-dependent promoters[23]. Regardless, additional target genes of SigR, were identified by RIViT-seq, thus expanding the SigR-mediated transcriptional regulatory network. Some target genes had possibly been previously unidentified because negative transcriptional regulation is involved in vivo. Of the SigR regulon identified by RIViT-seq, *SCO6404* was most highly expressed. This gene was not previously identified as a SigR target gene. While the function of SCO6404 is unknown, the downstream gene, *SCO6405*, encodes a DNA recombinase. The known regulon of SigR includes DNA repair proteins to cope with UV- and electrophile-incurred DNA damages[19]. DNA recombinase encoded by *SCO6405* may also be involved in this stress or similar response.

Mapping the 5′-ends of transcripts enabled prediction of the promoter sequences the sigma factors recognise. Because ShbA initiated the transcription from only two promoters, no significantly enriched motifs were found. When the 5′-ends of transcripts generated by SigR were compared, a consensus DNA sequence motif was modelled (Fig. 2). This motif resembled the previously identified SigR-dependent promoter sequences (GGAAT-N$_{18-19}$-GTT) including the 18 to 19 nucleotides spacer between and −10 and −35 regions[19]. Overall, RIViT-seq was able to identify genes directly transcribed by the RNA polymerase holoenzyme. In addition, the promoter sequences that the sigma factor recognised were inferred from RIViT-seq when a sufficient number of target genes were found and the promoter sequences were highly conserved.

**Application of RIViT-seq to uncharacterised sigma factors**. In order to better understand the transcriptional regulatory network in *S. coelicolor* A3(2), additional sigma factors were characterised by RIViT-seq. A total of 59 sigma factors that possess both Sigma70 region 2 and region 4 are encoded on the chromosome of *S. coelicolor* A3(2) (Supplementary Fig. 1). All the genes except for *SCO1723* were successfully cloned. Of them, 12 sigma factors including ShbA and SigR were successfully purified from soluble fractions of *E. coli* cell extracts (Supplementary Table 2). These sigma factors were HrdB, ShbA, SigF, SigH, and SigR, of which functions are known with at least one known target gene, SigI and SigK, which have been characterised by gene deletion and of which target genes are unknown, and SCO0864, SCO2742, SCO3626, SCO4895 and SCO6239, which were not previously characterised (Supplementary Table 1). Except for SCO3626, at least one target gene was identified for each sigma factor by RIViT-seq (Table 1; Supplementary Tables 3–13). The greatest number of target genes were identified for SigR followed SigI and SigH. For seven sigma factors, only one-three target gene(s) was identified. Notably, only one target gene was identified for HrdB although it is presumed to be responsible for transcriptional initiation of the majority of housekeeping genes. As activities of some sigma factors are controlled by posttranslational modification such as proteolysis or acetylation or additional factors such as RbpA, those sigma factors may be inactive or only partially functional without such modifications or factors[24–26]. Indeed, multiple forms of SigH, SigR, BldN and HrdB have been identified from cell extracts of *S. coelicolor* A3(2) and *S. griseus*[4,23,27,28].

**Modulation of sigma factor activities by proteolysis**. Because posttranslational modification plays a role in activating some sigma factors, the activities of sigma factors were further tested using protein variants presumed to mimic posttranslationally modified sigma factors. Some sigma factors possess an extra N- or C-terminal extension that prevents or limits RNA polymerase-binding or promoter recognition activity. Further analysis of protein domain organisations revealed that 17 sigma factors possessed an additional N- or C- terminal extension (> 100 amino acid residues). As the activity of these 17 sigma factors may be negatively affected by these extensions, genes encoding truncated

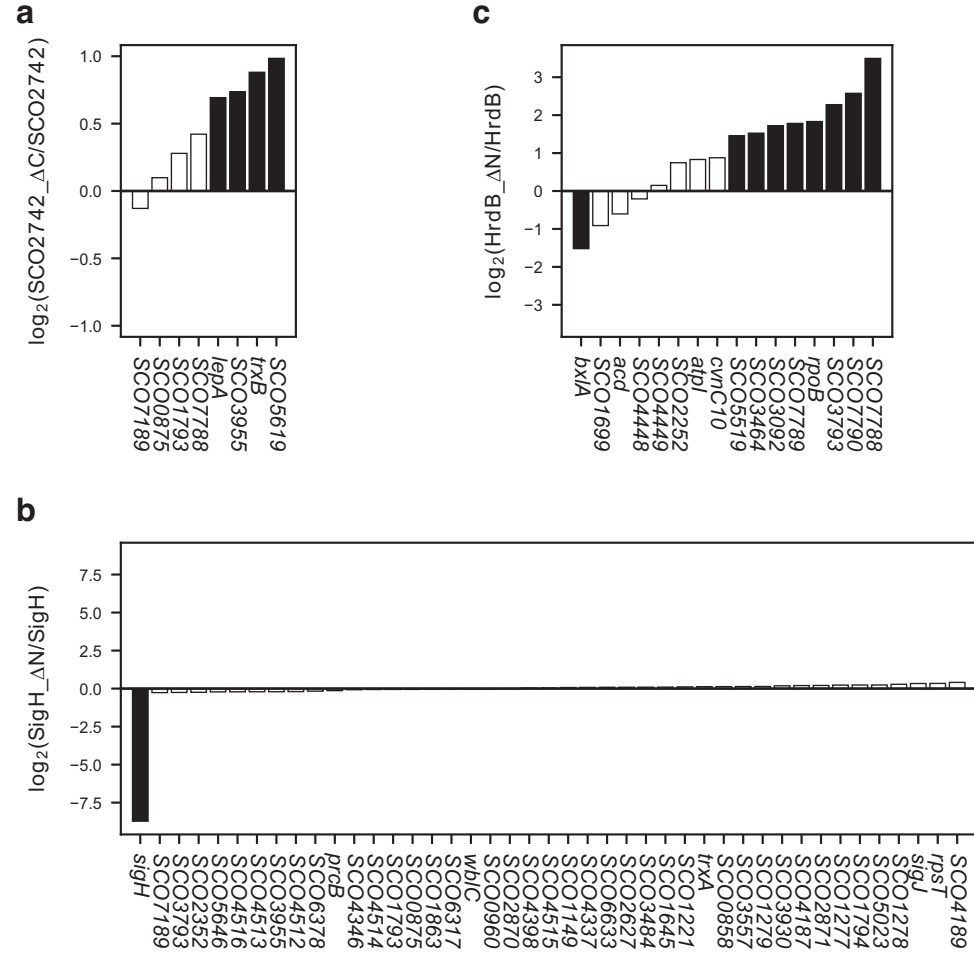

**Fig. 3 Relative transcript abundances generated by the truncated and full-length sigma factors.** *Y* axis indicates the relative abundance of the transcripts generated by the truncated sigma factor compared to the transcript abundance generated by the full-length form of the sigma factor. **a** Relative transcript abundances of the target genes of SCO2742. **b** Relative transcript abundances of the target genes of SigH. **c** Relative transcript abundances of the target genes of HrdB. Filled bars, adjusted *P* value < 0.01; open bars, adjusted *P* value ≥ 0.01. Fold changes and adjusted *P* values were determined by using the *DESeq2* package.

versions of these 17 sigma factors were cloned. Of them, 4 truncated sigma factors were successfully purified including three sigma factors of which full-length counterparts were also analysed by RIViT-seq (Supplementary Table 2). Using RIViT-seq, at least one target gene was identified for three truncated sigma factors (Table 1; Supplementary Tables 14–16). Greater numbers of genes were directly transcribed by the truncated versions of all three sigma factors than their full-length counterparts, suggesting that the N- or C-terminal extensions exert negative effects on RNA polymerase activity. The number of genes transcribed by SCO2742 substantially increased by removing the C-terminal extension. However, this modification only marginally (about 1.5 times) increased the transcriptional initiation activity of SCO2742 as judged by the fold change between SCO2742_ΔC and SCO2742, suggesting that the C-terminus of SCO2742 may be involved in only fine tuning the activity (Fig. 3a). Similarly, the activity of SigH only marginally changed by removing the N-terminal extension except for the *sigH* transcription (Fig. 3b). This suggests that the full-length form of SigH initiates the transcription of *sigH* actively while SigH is likely redirected to other promoters once its N-terminal extension is cleaved in order to promote transcription of other SigH-target genes. Unlike SCO2742 and SigH, N-terminal truncated HrdB substantially increased transcription initiation at multiple promoters, suggesting a negative regulatory role of the N-terminal extension

(Fig. 3c). Previously, two forms of HrdB were observed in *S. griseus* cells, indicating that HrdB could be partially processed[4]. Taken together, it is plausible that the activity of HrdB may be indeed modulated by proteolysis. In addition to the proteolysis, the activity of HrdB is known to be modulated by acetylation and RbpA[24,25]. When characterising transcription factors in cells, it is difficult to unambiguously confirm the effect of posttranslational modifications and cofactors. RIViT-seq enables systematic determination of their effect by using the modified proteins or supplementing the cofactors in the in vitro transcription assay.

**Expansion of the transcriptional regulatory network**. Prior to this study, the experimentally verified transcriptional regulatory network controlled by sigma factors in *S. coelicolor* A3(2) consisted of 12 sigma factors and 200 genes forming 209 sigma factor-target gene pairs or edges (Fig. 4; Supplementary Table 1)[4–8,10,15,18,19,25,29]. Our RIViT-seq data increased the number of sigma factors for which at least one biochemical target gene is known to 18 and the number of edges to 399 (Fig. 4). SigR and SigE control the transcription of more than 100 genes. The relatively small size of the HrdB regulon despite its function as the primary sigma factor is presumably because its activity is posttranslational controlled by acetylation and RbpA, which were not investigated in this study, in addition to proteolysis and only

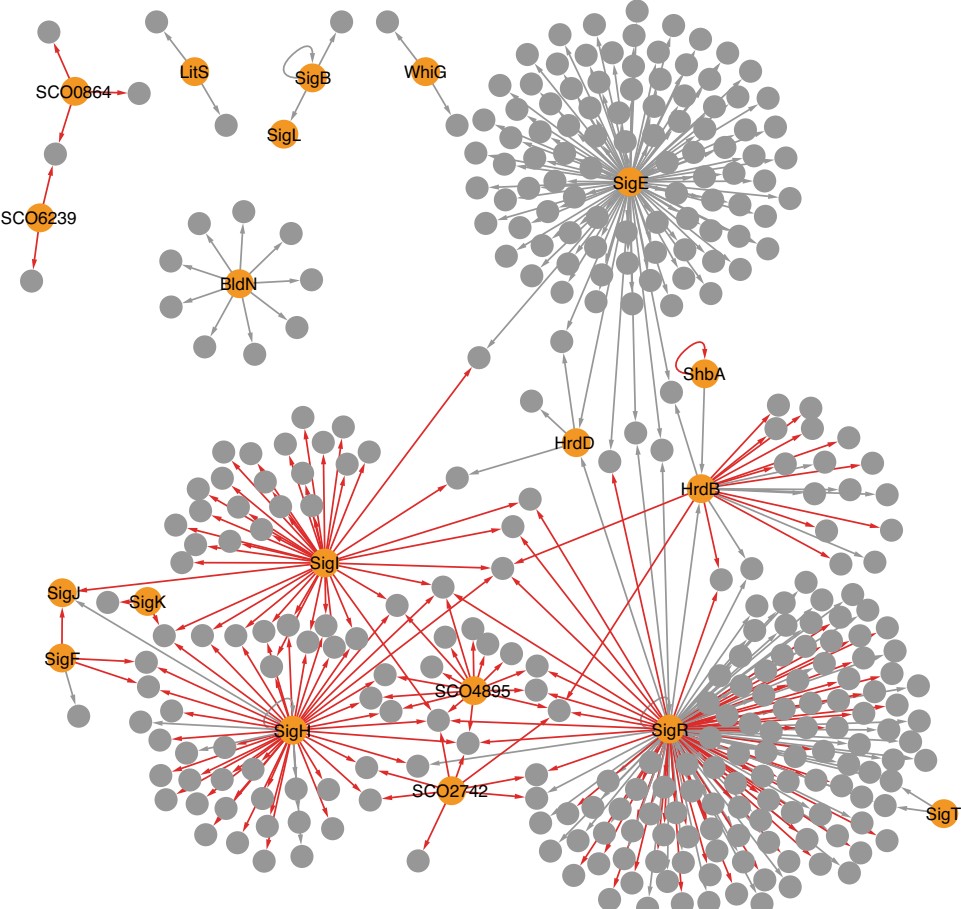

**Fig. 4 Transcriptional regulatory network controlled by sigma factors.** Orange nodes indicate sigma factors and grey nodes indicate other genes. Grey arrows are previously known regulation and red arrows are regulation identified by RIViT-seq in this study.

a limited number of target genes have been experimentally verified prior to this study. Four sigma factors initiate transcription of their own genes including *shbA*. Transcription of 4 sigma factor genes was initiated by other sigma factors. They are *hrdB* controlled by ShbA and SigR, *sigJ* controlled by SigH, SigI and SigF, *sigL* controlled by SigB, and *hrdD* controlled by SigR and SigE. Of them, the transcriptional dependence of *shbA* on ShbA and *sigJ* on SigI and SigF has only been determined by RIViT-seq, which needs to be further verified in vivo.

Identification of TSSs enabled the prediction of the recognition sequences of sigma factors. Presumably due to the relatively low promoter recognition specificity of some sigma factors and the low number of TSSs identified for several sigma factors, recognition motifs were modelled for only three sigma factors with high confidence ($E$ value $< 10^{-8}$; Fig. 2). This is consistent with the previous observation of low sequence conservation of DNA sequences bound by HrdB[30]. Interestingly, both SigH and SigI recognised the sequences rich in adenosine residues around their probable −10 regions, which may ease dsDNA melting similar to other sigma factors such as RpoD and RpoE in *E. coli*[1]. Their −35 regions were not highly conserved.

**Crosstalk between SigH, SigI, SigF and SigK**. The expanded transcriptional regulatory network also revealed that 45 out of 340 genes assigned to the sigma factor regulons could be directly regulated by multiple sigma factors (Fig. 4). Notably, *SCO1793* belonged to the regulons of five sigma factors and four genes (*SCO0875*, *SCO3793*, *SCO4515* and *SCO7788*) were regulated by four sigma factors. Of the 18 sigma factors of which regulons are

known including the previously identified target genes, SigH and SigI shared the greatest number of target genes in their regulons (Fig. 5a). However, not all the sigma factors recognise the same single promoter for the transcription of the same gene. For example, the *hrdB* transcription is initiated at two promoters, P1 and P2, with P1 recognised by ShbA and P2 recognised by SigR[4,19]. Therefore, the TSSs of transcripts detected by RIViT-seq were compared and the number of TSSs recognised by each of a pair of sigma factors was calculated (Fig. 5b). Similar to the regulons, SigH and SigI initiated transcription from the greatest number of the same TSSs in vitro. A comparison of their recognition motifs revealed two adenosine residues separated by two nucleotides at equivalent positions around their probable −10 regions (Fig. 2). Phylogenetic analysis of each of the sigma factor domains responsible for promoter recognition, region 2 and region 4, revealed that SigH and SigI belonged to the same clade, supporting their overlapping promoter recognition specificities (Supplementary Fig. 4).

Interestingly, the SigI regulon included an anti-anti-sigma factor gene, *bldG*. BldG antagonises the anti-SigH anti-sigma factor UshX, anti-SigF anti-sigma factor RsfA, and anti-SigK anti-sigma factor SCO7328[31–33]. RIViT-seq revealed that SigH, SigI and SigF initiated the transcription of *sigJ*, which is one of the known target genes of SigH, and SigH, SigI and SigK initiated the transcription of *wblC* (Fig. 4)[34]. Indeed, each of SigH, SigI and SigF is able to recognise the *ctc* promoter from *Bacillus subtilis*, routinely used promoter to probe stress response activities, suggesting they recognise similar promoter sequences[35,36]. Taken together, the expanded transcriptional regulatory network

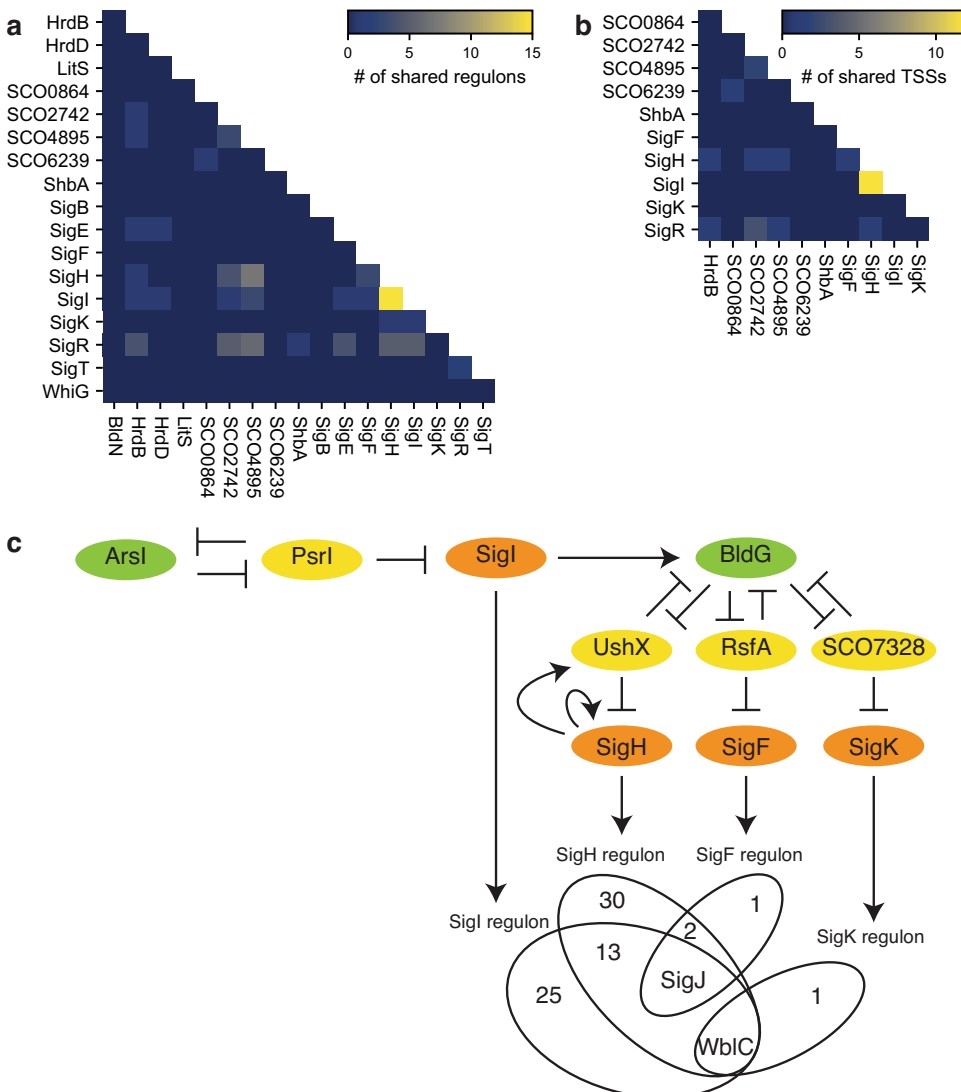

**Fig. 5 Crosstalk of sigma factors. a** Heatmap of the number of shared regulons between each pair of sigma factors. These regulons include the previously identified target genes. **b** Heatmap of the number of shared TSSs between each pair of sigma factors. Only TSSs identified by RIViT-seq were used. **c** Regulatory network controlled by SigH, SigI, SigF and SigK. Green, yellow and orange ovals indicate anti-anti-sigma factors, anti-sigma factors and sigma factors, respectively.

suggests that SigI promotes the expression of the SigH, SigF and SigK regulons by (i) activating the expression of *bldG* and (ii) directly initiating the transcription of some SigH, SigF and SigK regulons including *sigJ* and *wblC* (Fig. 5c).

## Discussion

Transcriptional regulation is a highly controlled process to ensure gene expression only under appropriate environmental conditions, and living organisms use a variety of transcription factors to activate or repress different sets of genes in response to environmental cues. Identifying target genes of a transcription factor is a laborious process, which typically includes transcriptomics analyses using knockout mutants and ChIP-seq, and is not scalable. Even a simple microorganism such as *E. coli* encodes hundreds of proteins and other types of regulatory elements predicted to be involved in transcriptional regulation and characterising each of them individually in vivo through extensive genetic manipulation is resource intensive. RIViT-seq presented in this study overcomes the issue of characterising a large number of proteins controlling the activities of the transcriptional machineries. Although the in vitro transcription technique was

previously combined with DNA microarray analysis (run-off transcription-microarray analysis, ROMA)[21,22,37,38], RIViT-seq overcomes several limitations ROMA has. Firstly, DNA microarray detects transcripts based on hybridisation between cDNA and probes and it is very difficult to rule out any non-specific hybridisation. Secondly, DNA microarray detects only transcripts that are complementary to probes arrayed on a chip. A high-density DNA microarray is required to detect multiple parts of a transcript and it's still not sufficient to detect structures of transcripts in a single nucleotide resolution unlike RNA-sequencing. RNA-sequencing enables determining the 5′- and 3′-ends of transcripts by creating sequencing libraries appropriately. Lastly, and presumably most importantly, ROMA requires a DNA microarray for every organism of interest.

In this study, RIViT-seq was successfully used to characterise sigma factors, determinants of promoter selectivity of RNA polymerase in bacteria, as a proof of concept. This technology is particularly useful if a transcription factor of interest undergoes negative regulation inside cells. For example, many sigma factors are sequestrated by their cognate anti-sigma factors from RNA polymerase and unable to express their regulons without specific stimuli[39].

In addition, transcriptional repressors and some nucleoid proteins block access of RNA polymerase to the promoter regions even if sigma factors are active until specific stimuli abrogate the DNA-binding activity of transcriptional repressors and nucleoid proteins. It is often a challenge to predict what kind of stimuli activate the sigma factor of interest and release the transcriptional repressor from its target promoter regions, and whether the activity is controlled by any anti-sigma factor and other transcription factors, which complicates its characterisation in vivo. In addition, indirect effects may exist when characterising a sigma factor and, even more broadly, any protein involved in transcriptional regulation in vivo. By contrast, in vitro transcription assays are a simpler method to characterise any proteins involved in transcriptional regulation. If the transcription factor of interest is predicted to undergo proteolysis, those modified transcription factors may be also characterised. In this study, we used truncated versions of four sigma factors which possess unusual N- or C-terminal extensions and observed the change in the transcription initiation activities. Sigma factor activity is sometimes further controlled by other factors such as RbpA and other types of post-translational modifications such as acetylation[24,25]. The effect of such additional factors and posttranslational modifications may be unambiguously determined by this in vitro-based technology on the genome scale.

There are, nevertheless, a few drawbacks of RIViT-seq and the procedure we used in this study. Most notable one is that all the components should be available for the in vitro transcription assay including the transcription factor and RNA polymerase. Not all the transcription factors are easily overproduced and purified as only 16 sigma factors were successfully purified in this study. The advancement in the in vitro transcription and translation technology and high throughput protein purification technology is expected to improve the success rate of the protein purification and enable the characterisation of a greater number of transcription factors by RIViT-seq. In the current procedure, we used RNA polymerase core enzyme from *E. coli*. Purifying RNA polymerase core enzyme is tedious work and requires equipment that not all research groups have access to. *E. coli* RNA polymerase core enzyme, which is commercially available, was successfully used with sigma factors from diverse bacteria including streptomycetes when conducting in vitro transcription assays previously[4,10–14]. There, however, may be cases in which interaction between organism-specific residues of RNA polymerase core enzyme and the sigma factor or promoter sequence is crucial. Hence, it may be desirable to prepare RNA polymerase core enzyme from the organism of choice to minimise potential false negatives when possible. The other possible drawback of the current procedure is the use of digested genomic DNA. The topological state of DNA is known to affect DNA binding of some transcription factors, thus influencing gene expression. Our current procedure of using relaxed DNA may overlook gene expression that requires a specific state of topology. Indeed, elevated chromosome supercoiling caused by topoisomerase I depletion changes global gene expression in *S. coelicolor* A3(2)[40]. In certain cases, it may be important or sometimes necessary to use undigested genomic DNA and add topoisomerase or gyrase to the in vitro transcription reaction.

Our data expanded the transcriptional regulatory network in *S. coelicolor* A3(2) (Fig. 4). Streptomycetes encode large numbers of sigma factors and adopt several unique regulatory mechanisms. For example, c-di-GMP controls the activity of WhiG, sporulation sigma factor, by binding the anti-WhiG anti-sigma factor, RsiG[5]. Another example of unique regulatory mechanisms is the control of the transcriptional initiation of the principal sigma factor gene, *hrdB*, by the alternative sigma factor, ShbA, under normal growth conditions[4]. However, how ShbA activity is controlled was previously unknown including which sigma factor is responsible for

the transcription of *shbA*. Our RIViT-seq data revealed that ShbA recognises the promoter of its own gene and initiates transcription in vitro. In addition to identifying the regulons of sigma factors, our data also revealed possible partial overlap of sigma factor regulons. The most notable case found in this study was partial overlap of the SigH, SigI, SigF and SigK regulons. *S. coelicolor* A3(2) encodes 10 sigma factors belonging to the group 3 based on their domain organisations and nine of them including SigH, SigI, SigF and SigK are considered homologues of the general stress response sigma factor, SigB, in *B. subtilis* (Supplementary Fig. 1)[41]. Similar to SigB of *B. subtilis*, the activities of SigH, SigI, SigF and SigK are controlled by their cognate anti-sigma factors and anti-anti-sigma factors[31,32,42]. The anti-SigH, anti-SigF and anti-SigK anti-sigma factors are antagonised by the anti-anti-sigma factor, BldG, and the anti-SigI anti-sigma factor, PsrI, is antagonised by the anti-anti-sigma factor, ArsI. RIViT-seq revealed that the *bldG* expression could be controlled by SigI. This regulatory cascade together with crosstalk may enable rapid response to multiple environmental changes where the SigH, SigI, SigF and SigK regulons need to be expressed. As the knockout mutants of these sigma factor genes exhibits mild or no phenotypes, these sigma factors may play partially complementary roles in *S. coelicolor* A3(2). It is still unknown what kind of stimuli directly control the activity of the anti-anti-sigma factors, BldG and ArsI, and how the *sigI* transcription is controlled. In *B. subtilis*, nine proteins are known to control the activity of the general stress response sigma factor, SigB[43]. Similarly, these SigB-like sigma factors in *S. coelicolor* A3(2) may also be controlled by multiple factors. Further investigation into the regulation of SigH, SigI, SigF and SigK should unveil this complex regulatory network that involves multiple sigma factors.

## Methods

**Bacterial strains and growth conditions**. *S. coelicolor* M145, derivative of *S. coelicolor* A3(2) lacking the plasmids SCP1 and SCP2, was used to isolate the genomic DNA[44]. *E. coli* TOP10 used for DNA cloning and *E. coli* BL21(DE3) used for protein overproduction were routinely cultivated in LB medium at 30 °C unless otherwise specified. Media were supplemented with kanamycin (25 mg/L) as appropriate.

**Bioinformatic analyses**. Pfam domain search of each predicted protein was performed using hpc_hmmsearch, modified hmmsearch from HMMER3 for efficient use of CPU cores on the Cori supercomputer, with a threshold of independent $E$ value of 0.1[45,46]. Proteins predicted to possess both Pfam domains Sigma70_r2 (region 2) and Sigma70_r4 (region 4) were considered sigma factors. PhyML was used to estimate the phylogeny by maximum likelihood[47]. MEME was used to find consensus motifs[48]. Motifs that consisted of 22 to 35 nucleotides and present in at least 10 sequences with the $E$ values $< 10^{-5}$ were retrieved. The motif with the smallest $E$ value was used as the consensus sequence.

**DNA cloning**. Sigma factor genes were amplified by PCR using KOD Hot Start DNA Polymerase (MilliporeSigma) and primer pairs listed in Supplementary Table 2 and amplicons were cloned by Gibson Assembly (New England BioLabs) into pET29b(+) digested by *Nde*I and *Xho*I.

**Purification of hexahistidine-tagged sigma factors**. *E. coli* possessing the pET29b(+)-derived plasmid was cultivated in 2 ml LB medium containing 25 μg/ml kanamycin overnight at 30 °C. To a fresh 50 ml LB medium containing 25 μg/ml kanamycin, 1 ml of the preculture was inoculated. The cells were grown at 30 °C for 2 h and the temperature was changed to 15 °C. IPTG was added to the final concentration of 0.1 mM for SCO5243 or 1 mM for all other sigma factors and the cultivation was continued overnight. Cells were harvested by centrifugation and stored at –80 °C until the use. Cell pellets were resuspended in a buffer containing 20 mM HEPES (pH 7.5), 50 mM sodium chloride, >0.25 units/μl Benzonase Nuclease (MilliporeSigma), 1 mM magnesium sulphate and 1 × BugBuster (MilliporeSigma), and lysed for 30 min at room temperature with continuous agitation. Insoluble materials were removed by centrifugation and imidazole was added to the soluble fraction at 20 mM. The soluble fraction was mixed with HisPur Cobalt (Thermo Scientific) and incubated for 2 h with continuous agitation. The unbound materials were removed and the resin was washed with a buffer containing 20 mM HEPES (pH 7.5), 50 mM sodium chloride and 30 mM imidazole. The bound protein was eluted with a buffer containing 20 mM HEPES (pH 7.5), 50 mM sodium chloride and 150 mM imidazole. Imidazole was removed by using Microcon-10kDa

Centrifugal Filter Unite (MilliporeSigma). The protein concentration was measured using Bio-Rad Protein Assay with BSA as the titration standard (Bio-Rad).

**In vitro transcription assay**. The genomic DNA of *S. coelicolor* A3(2) was isolated by using GenElute Bacterial Genomic DNA kit (MilliporeSigma) and digested by *Eco*RI, *Hin*dIII, *Bam*HI or *Xho*I. Equal quantities of the genomic DNA solutions treated by different restriction enzymes were combined. In vitro transcription assays were performed using Echo® 525 LIQUID HANDLER (Labcyte). Two μl of 8 μM sigma factor, 1 μl of one unit/μl *E. coli* RNA polymerase core enzyme (New England BioLabs) and 2 μl of 5X *E. coli* RNA polymerase reaction buffer were mixed and incubated at 30 °C for 15 min. To this mixture, 1 μl of 200 ng/μl digested genomic DNA mixture was added and the mixture was incubated at 30 °C for 15 min. The in vitro transcription reaction was initiated by adding 2 μl of 2.5 mM NTP mixture (Invitrogen) and the mixture was incubated at 30 °C for 20 min. To the reaction was the DNase solution consisting of 2 μl TURBO DNase (2 units/μl; Ambion), 1.5 μl of 10X TURBO DNase buffer and 1 μl of ERCC RNA Spike-in Mix (Invitrogen) diluted by 100 times added. The reaction was incubated at 37 °C for 30 min and the RNAs were purified using RNeasy kit (Qiagen). RNA concentrations were quantified using Qubit™ RNA HS Assay Kit (Invitrogen). The cDNA synthesis was performed using the SuperScript IV First-Strand Synthesis System (Invitrogen) and the quantitative RT-PCR was performed using SsoAdvanced Universal SYBR Green Supermix (Bio-Rad Laboratories) and primer pairs listed in Supplementary Table 3 on CFX384 Touch Real-Time PCR System (Bio-Rad Laboratories).

**Whole-genome transcriptomics**. Stranded cDNA libraries were generated using TruSeq Stranded RNA Library Prep Kit (Illumina) and the low sample protocol. Libraries were quantified using KAPA Library Quantification Kit (Roche) and LightCycler 480 Instrument (Roche), and sequenced on NovaSeq 6000 Sequencing System (Illumina) by paired-end 2 × 150 bp sequencing.

Raw reads were scanned from 3′ to 5′ and those with a quality score value below 20 were trimmed and reads consisting of fewer than 35 nucleotides were discarded using BBDuk (sourceforge.net/projects/bbmap/). Trimmed reads were aligned to the *S. coelicolor* A3(2) chromosome and ERCC92 sequences using HISAT2 with the "no-spliced-alignment" option and the "maxins" option of 1,000[49]. The number of fragments overlapping each gene was counted using featureCounts[50]. Fragment counts were normalised by ERCC fragment counts using the R package *RUVSeq* and differential expression was analysed using the R package *DESeq2*[51,52]. The fold change of ≥2 and the adjusted *P* value of <0.01 were used to determine the significantly overexpressed genes.

**5′-end sequencing**. The triphosphates at the 5′-end of 6 μg RNA samples were converted to monophosphates by using 3.75 units of RNA 5′ pyrophosphohydrolase (RppH; New England BioLabs) at 30 °C for 1 h. The 5′-ends of the transcripts were ligated by the adaptor (GUUCAGAGUUCUACA-GUCCGACGAUC) using 15 units of T4 RNA ligase 1 (ssRNA ligase; New England BioLabs) at 25 °C for 2 h followed by incubation at 17 °C for 18 h. Ligated RNA samples were purified by using RNeasy Kits and RNeasy MinElute Cleanup Kits (QIAGEN). cDNA was synthesised by using a random hexamer with an adaptor sequence (GCCTTGGCACCCGAGAATTCCANNNNNNN) and SuperScript IV First-Strand Synthesis System (Invitrogen). The cDNA was amplified by using KAPA Library Amplification Kit (Roche) and index primers from TruSeq Small RNA Library Preparation Kit (Illumina). Libraries were purified by using the equal volume of AMPure XP (Beckman Coulter) twice. The quality of the libraries was verified by using Bioanalyzer High Sensitivity DNA Kit (Agilent). The libraries were sequenced on MiSeq System (Illumina) by paired-end 2 × 150 bp sequencing.

Adaptor sequences present at the 3′-end of reads were trimmed using BBDuk (sourceforge.net/projects/bbmap/). Raw reads were scanned from 3′ to 5′ and those with a quality score value below 20 were trimmed and reads consisting of fewer than 35 nucleotides were discarded using BBDuk (sourceforge.net/projects/bbmap/). Trimmed reads were aligned to the *S. coelicolor* A3(2) chromosome and ERCC92 sequences using HISAT2 with the "no-spliced-alignment" option and the "maxins" option of 1000[49]. The number of 5′-end of forward reads that aligned each genomic position was counted using Samtools[53]. Transcriptional start sites (TSSs) were determined using the procedure reported previously (Supplementary Fig. 5a)[16]. Briefly, 5′-ends within 100 bp were clustered together. If multiple 5′-ends located nearby had standard deviation of <10, they were subclustered together and the 5′-end that had the largest read count within the subcluster was considered the TSS. The sum of the read counts of all the 5′-ends within the subcluster was used as the read count of the TSS. ERCC92 transcripts that had at least 10 reads aligning the 1st nucleotide in all the replicates were used as a normalisation control. The read count relative to the maximum number among samples was calculated for each ERCC92 transcript and the mean value of all the relative read counts was used as the normalisation factor, which ranges between 0 and 1 (Supplementary Fig. 5b). TSSs with the normalised read counts (read counts divided by the mean normalisation factor) of <4 in at least one of the replicates in the sample with a sigma factor were removed from the further analysis. As 5′-end mapping data were skewed, the following criteria were used to determine sigma factor-dependent TSSs. (i) The normalised read counts in every replicate of the sample with a sigma factor is greater than the read count in all the replicates of the sample with no sigma factor. (ii) The normalised read count in the sample with a sigma factor with the 2nd lowest normalised read count is at least four times as much as the read count in all the replicates of the sample with no sigma factor.

**Integration of the whole transcriptomics and 5′-end sequencing data**. The significantly overexpressed genes in the whole transcriptomics data were further analysed for the presence of the TSSs. Overexpressed genes that had a TSS within 300 bp upstream from their initiation codon were determined the target genes of the sigma factor. If an overexpressed gene with no TSS was located within 50 bp downstream from a target gene, it was also determined a target gene.

**Reporting summary**. Further information on research design is available in the Nature Research Reporting Summary linked to this article.

## Data availability

All sequence files and processed count data files that support this study are available at Gene Expression Omnibus (GEO) under accession number GSE184392 (whole transcriptome) and GSE184393 (5′-end sequencing). Source data are provided with this paper.

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

## Acknowledgements

This work was supported by the US Department of Energy Office of Science under Contract No. DE-AC02-05CH11231.

## Author contributions

H.O. designed and executed the experiments, conducted the data analysis and wrote the manuscript. N.J.M. supervised the work and wrote the manuscript.

## Competing interests

The authors declare no competing interests.
