## [Peer Review File · Nature Communications]

REVIEWER COMMENTS

Reviewer #1 (Remarks to the Author):

Otani and Mouncey describe use of *in vitro* transcription of fragmented *Streptomyces* chromosomal DNA as a way to identify genes and promoters whose transcription is regulated by different sigma factors. Their work is an advance of prior, similar approaches by use of transcription start site (TSS) sequencing methods and modern RNA-seq in place of microarrays. These sequencing methods enable a significant step forward in this strategy and as such generate results that in principle are appropriate for high-profile publication. The authors attempt to test all 57 known sigma factors or candidate sigma factors from the *Streptomyces* strain selected, but somewhat surprisingly were only able to generate results for 11. This inefficiency may reflect a number of limitations of the method in addition to limited efforts at expressing soluble forms of the sigma factors for purification. Nonetheless, the authors were able to generate sufficient results to validate the method and to make some contributions to better understanding of the transcriptional regulatory network of the *Streptomyces* species. Several aspects of the work, enumerated below, are suboptimal. These issues were incompletely addressed in the initial submission and will require revision of the manuscript before it is acceptable for publication.

Major issues

1. A key point in the authors' assay that they do not highlight in the manuscript and only mention in the supplemental methods is that they used *E. coli* RNA polymerase to map *Streptomyces* promoters with *Streptomyces* sigma factors. Although sigma factors do often work with RNA polymerases from different bacterial species, this is not universally the case. The greater the evolutionary distance between the species, the greater the concern will be that lineage-specific interactions between the sigma and its cognate RNA polymerase may prevent the sigma factor from functioning properly. Further, some promoter sequences contact the core RNA polymerase and not the sigma factor. Although such interactions appear to be largely auxiliary rather than determinative, they nonetheless may be needed to generate sufficient transcript for detection. There is no reason, other than convenience, that the authors could not do the same experiments using *Streptomyces* RNA polymerase. It would likely improve the results of the method. The authors need to mention the use of *E. coli* RNA polymerase when the method is introduced in the Results. Additionally, they need to discuss the limitations its use introduces to their method (see point 3 below).

2. About 200 new gene targets for *Streptomyces* sigma factors are identified by the *in vitro* transcription experiments reported, including targets for several sigma factors for which no target was previously known. The authors state these results as definitive conclusions. However, no experiments using either all *Streptomyces* components *in vitro* or genetic experiments *in vivo* are included to confirm these assignments. Confirming results, e.g. by quantitative PCR methods or other means, is standard practice in most genome-scale sequencing work. In lieu of such confirmatory experiments, the authors should report their findings as proposed sigma-gene relationships rather than as definitive conclusions as they are currently stated. For example, at line 253 the authors write that their findings "...revealed that SigI promotes the expression of the SigH, SigF, and SigK regulons...." This should instead be written and "...suggest that SigI promotes...." This same type of revision should be made throughout the manuscript where conclusions are drawn. In lieu of confirming results using *Streptomyces* RNA polymerase or by some *in vivo* experiment, the conclusions must be regarded as tentative. This said, of course, the manuscript would be significantly strengthened if the authors could confirm one or more of the tentative assignments with some type of follow-on experiment.

3. Another major revision that is need prior to acceptance of this manuscript is the addition of an explicit and discrete section to the discussion that describes the limitations to the method and possible ways to overcome time. The use of non-cognate RNA polymerase is one example of a limitation that must be addressed in this manner. In the results, the authors mention that the absence of activators and co-factors (eg, RpbA) in their assay is another limitation. This limitation should be discussed more fully in the discussion section, which will afford an opportunity to describe the utility of the adding such factors to the assay. The authors might also point out that small molecule regulators like ppGpp are known to regulate promoters in some bacteria. Their assay could be a way to study such effects on the genome scale.

Additionally, the authors' method has a significant limitation, which they don't mention, in using only relaxed rather than negatively supercoiled DNA. Bacterial chromosomes are generally negatively supercoiled. This supercoiling can be modulated and in some cases increased at promoters. Negative supercoiling aids DNA melting at promoters. Thus, promoters in relaxed DNA are typically less active that they are in negatively supercoiled DNA. The role of supercoiling in promoter function in Streptomyces is largely uncharacterized. Thus, it is unclear if some Streptomyces sigma factors might rely on negative supercoiling for transcription initiation. The authors should discuss this limitation. They may also wish to mention possible approaches to address it.

Minor points

- 1. line 37, "if not only" is redundant and can be deleted.**
- 2. line 47, "proteins domains" should be just "domains"**
- 3. line 50, consider just "or" in place of the ungrammatical "and/or".**
- 4. line 65-67. This is a run-on sentence. Consider breaking up. Alternatively, replace "and successfully" with ", successfully" and add a comma before "and expanded".**
- 5. line 97. The numbers reported (μM) are concentrations not copy numbers. Please correct. Also, are these numbers correct? 15 mM (15,000 μM) is a very high concentration for an RNA transcript, equivalent to 15 million transcripts in a cell. It seems unlikely such concentrations are achieved in the in vitro assay.**
- 6. line 107. It seems like an example of the TSS data should be shown in Figure 1 or somewhere. Readers would benefit from a graphic illustration of the definition of TSSs from actual data.**
- 7. line 142-151. Was variable spacing between the -10 and -35 elements considered in searching for consensus sequences? Variation in the spacing of these promoter elements is well known, and methods have been developed to consider variable spacing in defining promoters have been developed (eg, see Shultzaberger et al. 2007. NAR 35:771-788).**
- 8. paragraph preceding line 204. What is the actual evidence that sigma factors in Streptomyces are proteolytically cleaved? I looked at reference 4, but proteolysis was not mentioned in the text of that manuscript (or at least did not appear in a text search for "proteolysis", "proteolytic", or "protease"). Is this idea just speculation or is there documented evidence for precursor forms of sigma factors in Streptomyces? It is well known that auxiliary domains of sigma factors can be auto-inhibitory for sigma factor function(eg, E. coli sigma70 region 1.1). Such domains on a Streptomyces sigma factor could fail to be properly activated by E. coli RNA polymerase. The authors should consider**

and this discuss these points.

9. line 286. When discussing the advantages of the in vitro method, the authors may wish to also mention that repressors can block promoter function and that such repressors would be absent in vitro.

10. line 295. It seems odd to mention that RNA polymerase must be available for the assay and not mention that it would be preferable to use the cognate RNA polymerase.

Reviewer #2 (Remarks to the Author):

In this manuscript, Otani et al. applied Illumina RNA seq to a new target – genome-wide in vitro transcription reactions, which they named RIViTseq. The authors demonstrated the utility of their approach by focusing on sigma factors from the organism *Streptomyces coelicolor*, a model species within the genus known to harbour a large number (>50) sigma factors, most of which are of the 'extra cytoplasmic function' variety. The authors optimised the concentration various reagents important in their assays using two previously experimentally characterised sigma factors prior to completing an *E. coli* protein production trial for all 59 sigma factors encoded by *S. coelicolor*. The authors then performed RIViTeq with the 13 sigma factors produced in a soluble form by *E. coli*, as well as a handful of truncated variants of those whose N-terminal domain was hypothesised to play a regulatory role. In total, the authors identified/extended the regulons for 11 sigma factors and demonstrated several new points of connectivity/redundancy in the resultant regulatory network. The application of RNA sequencing to in vitro transcription reactions that utilise the whole genome as a template DNA is potentially a very powerful technique that could address current weaknesses with global transcriptomic approaches. The authors were able to identify both previously known and new members of regulons in their study and TSSs identified were within 2nt of those previously published, validating the approach is producing accurate information at the nucleotide level.

Overall, I enjoyed reading this study. It was well designed and presented with the exception of using numbers in the text instead of words (i.e. spellout numbers less than 10). The high background problem cause by apo-RNAP was negated appropriately and statistical analyses appear to have done correctly. I have no experimental requests.

REVIEWER COMMENTS

Thank you for reviewing our manuscript and providing a number of comments, which helped improve our manuscript. We revised our manuscript based on reviewers' comments, which are highlighted in red. Our point-by-point response to reviewers' comments is provided below. We also revised bar charts in figure 1 and supplementary figures 2 and 3 to include individual data points in addition to the average and standard deviation, and the wording in accordance with Nature Communications' formatting guideline. We moved all the supplementary tables to the supplementary information file as these tables are not too large.

Reviewer #1 (Remarks to the Author):

Otani and Mouncey describe use of *in vitro* transcription of fragmented *Streptomyces* chromosomal DNA as a way to identify genes and promoters whose transcription is regulated by different sigma factors. Their work is an advance of prior, similar approaches by use of transcription start site (TSS) sequencing methods and modern RNA-seq in place of microarrays. These sequencing methods enable a significant step forward in this strategy and as such generate results that in principle are appropriate for high-profile publication. The authors attempt to test all 57 known sigma factors or candidate sigma factors from the *Streptomyces* strain selected, but somewhat surprisingly were only able to generate results for 11. This inefficiency may reflect a number of limitations of the method in addition to limited efforts at expressing soluble forms of the sigma factors for purification. Nonetheless, the authors were able to generate sufficient results to validate the method and to make some contributions to better understanding of the transcriptional regulatory network of the *Streptomyces* species. Several aspects of the work, enumerated below, are suboptimal. These issues were incompletely addressed in the initial submission and will require revision of the manuscript before it is acceptable for publication.

Major issues

1. A key point in the authors' assay that they do not highlight in the manuscript and only mention in the supplemental methods is that they used *E. coli* RNA polymerase to map *Streptomyces* promoters with *Streptomyces* sigma factors. Although sigma factors do often work with RNA polymerases from different bacterial species, this is not universally the case. The greater the evolutionary distance between the species, the greater the concern will be that lineage-specific interactions between the sigma and its cognate RNA polymerase may prevent the sigma factor from functioning properly. Further, some promoter sequences contact the core RNA polymerase and not the sigma factor. Although such interactions appear to be largely auxiliary rather than determinative, they nonetheless may be needed to generate sufficient transcript for detection. There is no reason, other than convenience, that the authors could not do the same experiments using *Streptomyces* RNA polymerase. It would likely improve the results of the method.

The authors need to mention the use of *E. coli* RNA polymerase when the method is introduced in the Results. Additionally, they need to discuss the limitations its use introduces to their method (see point 3 below).

Response. We agree that using *Streptomyces* RNA polymerase core enzyme may improve the result of the assay. Indeed, we attempted to purify the individual components of *Streptomyces* RNA polymerase core enzyme (α , β , β' and ω subunits). But not all of the subunits were overproduced in soluble forms in *E. coli*, and optimising the overproduction and purification

procedure appeared to be challenging. Purification of RNA polymerase core enzyme complex from streptomycetes (and likely other organisms) is multistep purification using an FPLC system with multiple columns, some of which are not commonly used. Other researchers wishing to perform RIViT-seq to characterise sigma factors from their organisms of choice may face the same issue. The primary objective of this study is demonstrating the RIViT-seq technology. Hence, we used *E. coli* RNA polymerase core enzyme, which is commercially available and other researchers may also use, to establish this new technology. *E. coli* RNA polymerase core enzyme was previously used with sigma factors from diverse bacteria for *in vitro* transcription assays including *Streptomyces* sigma factors.

In the revised manuscript, we state that we used *E. coli* RNA polymerase core enzyme and cited several papers in which *E. coli* RNA polymerase core enzyme was successfully used for *in vitro* transcription assays with sigma factors from a variety of bacteria. Notably, one study demonstrated that *E. coli* RNA polymerase core enzyme exhibited similar activity to mycobacterial RNA polymerase core enzyme, supporting use of *E. coli* RNA polymerase core enzyme with sigma factor from different bacteria for *in vitro* transcription assays. We also revised the discussion section and incorporated some of the possible limitations our current protocol may have.

Hu, Y., Morichaud, Z., Chen, S., Leonetti, J. P. & Brodolin, K. Mycobacterium tuberculosis RbpA protein is a new type of transcriptional activator that stabilizes the sigma A-containing RNA polymerase holoenzyme. *Nucleic Acids Res* **40**, 6547-6557, doi:10.1093/nar/gks346 (2012).

2. About 200 new gene targets for *Streptomyces* sigma factors are identified by the *in vitro* transcription experiments reported, including targets for several sigma factors for which no target was previously known. The authors state these results as definitive conclusions. However, no experiments using either all *Streptomyces* components *in vitro* or genetic experiments *in vivo* are included to confirm these assignments. Confirming results, e.g. by quantitative PCR methods or other means, is standard practice in most genome-scale sequencing work. In lieu of such confirmatory experiments, the authors should report their findings as proposed sigma–gene relationships rather than as definitive conclusions as they are currently stated. For example, at line 253 the authors write that their findings “...revealed that SigI promotes the expression of the SigH, SigF, and SigK regulons...” This should instead be written and “...suggest that SigI promotes...” This same type of revision should be made throughout the manuscript where conclusions are drawn. In lieu of confirming results using *Streptomyces* RNA polymerase or by some *in vivo* experiment, the conclusions must be regarded as tentative.

This said, of course, the manuscript would be significantly strengthened if the authors could confirm one or more of the tentative assignments with some type of follow-on experiment.

Response. We tried to confirm some of the new sigma factor-target gene assignments using knock down mutants, but we were unable to confirm them unambiguously. Many sigma factors are known to be activated under specific conditions and it is difficult to predict under which conditions a given sigma factor becomes active. For example, the activity of SigI, one of the sigma factors highlighted in this study, is controlled by its cognate anti-sigma factor and anti-anti-sigma factor. The knockout mutant of the *sigI* gene exhibited no phenotypic difference in the previous study. It is likely that the growth condition we used for confirmation of the new assignments was not where these sigma factors are active. This challenge is indeed one of the advantages of using RIViT-seq as we are able to identify target genes of transcription factors

including sigma factors without knowing the growth conditions to test. Nevertheless, we agree that the new assignments are not confirmed *in vivo* nor using all *Streptomyces* components. We revised the **Expansion of the transcriptional regulatory network and Crosstalk between SigH, SigI, SigF and SigK** sections of **Results** and the last paragraph of **Discussion** to make it clearer that the new assignments are solely based on the biochemical reactions *in vitro*.

Homerova, D., Sevcikova, B., Rezuchova, B. & Kormanec, J. Regulation of an alternative sigma factor sigmaI by a partner switching mechanism with an anti-sigma factor PrsI and an anti-anti-sigma factor ArsI in *Streptomyces coelicolor* A3(2). *Gene* **492**, 71-80, doi:10.1016/j.gene.2011.11.011 (2012).

3. Another major revision that is need prior to acceptance of this manuscript is the addition of an explicit and discrete section to the discussion that describes the limitations to the method and possible ways to overcome time. The use of non-cognate RNA polymerase is one example of a limitation that must be addressed in this manner. In the results, the authors mention that the absence of activators and co-factors (eg, RpbA) in their assay is another limitation. This limitation should be discussed more fully in the discussion section, which will afford an opportunity to describe the utility of the adding such factors to the assay. The authors might also point out that small molecule regulators like ppGpp are known to regulate promoters in some bacteria. Their assay could be a way to study such effects on the genome scale.

Additionally, the authors' method has a significant limitation, which they don't mention, in using only relaxed rather than negatively supercoiled DNA. Bacterial chromosomes are generally negatively supercoiled. This supercoiling can be modulated and in some cases increased at promoters. Negative supercoiling aids DNA melting at promoters. Thus, promoters in relaxed DNA are typically less active that they are in negatively supercoiled DNA. The role of supercoiling in promoter function in *Streptomyces* is largely uncharacterized. Thus, it is unclear if some *Streptomyces* sigma factors might rely on negative supercoiling for transcription initiation. The authors should discuss this limitation. They may also wish to mention possible approaches to address it.

Response. We added a new paragraph in the discussion section to introduce several drawbacks or limitations of the current procedure of RIViT-seq we present and possible solutions. Lack of certain additional factors is mentioned in the preceding paragraph about the positive aspect of RIViT-seq of the revised manuscript because this *in vitro*-based technology enables determining the effect of those additional factors unambiguously by adding them to the *in vitro* transcription reaction.

Minor points

1. line 37, "if not only" is redundant and can be deleted.

Response. Done

2. line 47, "proteins domains" should be just "domains"

Response. Done

3. line 50, consider just “or” in place of the ungrammatical “and/or”.

Response. Done

4. line 65-67. This is a run-on sentence. Consider breaking up. Alternatively, replace “and successfully” with “, successfully” and add a comma before “and expanded”.

Response. Done

5. line 97. The numbers reported (μM) are concentrations not copy numbers. Please correct. Also, are these numbers correct? 15 mM (15,000 μM) is a very high concentration for an RNA transcript, equivalent to 15 million transcripts in a cell. It seems unlikely such concentrations are achieved in the *in vitro* assay.

Response. This is a typo, it should be pM. Also, we realised that the concentrations originally presented were those in the stock solution, not in the actual samples (diluted by 100 times prior to addition). In the revised manuscript, we recalculated the quantity, which is presented using attomole, the unit the supplier uses.

The ERCC RNA Spike-in Mix purchased from Invitrogen was added as normalisation controls and these transcripts are not products of the *in vitro* reaction.

6. line 107. It seems like an example of the TSS data should be shown in Figure 1 or somewhere. Readers would benefit from a graphic illustration of the definition of TSSs from actual data.

Response. The TSS determination process is illustrated in Supplementary Figure 5, which readers may refer to. Because this is a multistep process, it is difficult to present the actual 5'-end sequencing data in a single figure. Instead, we show the TSSs determined by 5'-end sequencing together with the previously determined TSSs in Figure 1d of the revised manuscript.

7. line 142-151. Was variable spacing between the -10 and -35 elements considered in searching for consensus sequences? Variation in the spacing of these promoter elements is well known, and methods have been developed to consider variable spacing in defining promoters have been developed (eg, see Shultzaberger et al. 2007. NAR 35:771–788).

Response. Using different algorithms may improve promoter prediction, especially because the distance between the -10 and -35 elements is not always exactly the same. However, we first conducted the consensus motif analysis of the SigR-target promoters to demonstrate that the motif generated by using the RIViT-seq data is similar to the previously identified SigR-recognition motif and the RIViT-seq data are consistent with previous reports. The previous study only used WebLogo and did not describe how exactly those sites were chosen. Nevertheless, WebLogo cannot take account for the variability of spacing and it is very likely that the majority of the sequences used in the previous study had 19 nt spacing (the figure has 19 nt between the -10 and -35 elements; see the reference below). Therefore, we used the algorithm, MEME, widely used motif discovery tool which does not take account for the variability of the spacing, to identify a consensus motif. We then continued using the same algorithm to find motifs recognised by other sigma factors for consistency. Although the algorithm suggested by the reviewer may possibly improve the prediction, the main purpose of

conducting this analysis in this manuscript is demonstrating how the RIViT-seq data can be used rather than how they should be processed, and we think comparing and testing multiple algorithms for the promoter prediction is outside of the scope of this manuscript.

Kim, M. S. *et al.* Conservation of thiol-oxidative stress responses regulated by SigR orthologues in actinomycetes. *Mol Microbiol* **85**, 326-344, doi:10.1111/j.1365-2958.2012.08115.x (2012).

8. paragraph preceding line 204. What is the actual evidence that sigma factors in *Streptomyces* are proteolytically cleaved? I looked at reference 4, but proteolysis was not mentioned in the text of that manuscript (or at least did not appear in a text search for “proteolysis”, “proteolytic”, or “protease”). Is this idea just speculation or is there documented evidence for precursor forms of sigma factors in *Streptomyces*? It is well known that auxiliary domains of sigma factors can be auto-inhibitory for sigma factor function (eg, *E. coli* sigma70 region 1.1). Such domains on a *Streptomyces* sigma factor could fail to be properly activated by *E. coli* RNA polymerase. The authors should consider and this discuss these points.

Response. In the reference 4, two bands of HrdB were detected by western blotting and the authors noted that “HrdB could be partially processed; degradation of its N- or C-terminal portion is most likely”. Although there is currently no direct evidence of HrdB being “proteolysed”, it was demonstrated that two forms of HrdB exist in cells. While HrdB is the principal sigma factor in streptomycetes, its N-terminus sequence is not similar to Pfam domain of Sigma70 region 1.1 and the role of the N-terminus extension of HrdB is hardly known yet. We modified the text to better reflect what is known and what was suggested based on previous data.

9. line 286. When discussing the advantages of the in vitro method, the authors may wish to also mention that repressors can block promoter function and that such repressors would be absent *in vitro*.

Response. This is a great suggestion. We incorporated it in **Discussion**.

10. line 295. It seems odd to mention that RNA polymerase must be available for the assay and not mention that it would be preferable to use the cognate RNA polymerase.

Response. We revised this section and discussed use of RNA polymerase from *E. coli* and the organism of choice in the revised manuscript.

Reviewer #2 (Remarks to the Author):

In this manuscript, Otani et al. applied Illumina RNA seq to a new target – genome-wide in vitro transcription reactions, which they named RIViTseq. The authors demonstrated the utility of their approach by focusing on sigma factors from the organism *Streptomyces coelicolor*, a model species within the genus known to harbour a large number (>50) sigma factors, most of which are of the ‘extra cytoplasmic function’ variety. The authors optimised the concentration various reagents important in their assays using two previously experimentally characterised sigma factors prior to completing an *E. coli* protein production trial for all 59 sigma factors encoded by *S. coelicolor*. The authors then performed RIViTeq with the 13 sigma factors produced in a soluble form by *E. coli*, as well as a handful of truncated variants of those whose N-terminal

domain was hypothesised to play a regulatory role. In total, the authors identified/extended the regulons for 11 sigma factors and demonstrated several new points of connectivity/redundancy in the resultant regulatory network. The application of RNA sequencing to *in vitro* transcription reactions that utilise the whole genome as a template DNA is potentially a very powerful technique that could address current weaknesses with global transcriptomic approaches. The authors were able to identify both previously known and new members of regulons in their study and TSSs identified were within 2nt of those previously published, validating the approach is producing accurate information at the nucleotide level.

Overall, I enjoyed reading this study. It was well designed and presented with the exception of using numbers in the text instead of words (i.e. spellout numbers less than 10). The high background problem cause by apo-RNAP was negated appropriately and statistical analyses appear to have done correctly. I have no experimental requests.

Response. We appreciate reviewer's very positive comments on our study and manuscript. We spelled out numbers accordingly.

REVIEWERS' COMMENTS

Reviewer #1 (Remarks to the Author):

The authors have done a reasonable job of responding to the reviewers' critiques. Although some revisions that might have increased the impact of the paper were not implemented, overall the work represents a solid advance. The revisions have resolved my main concerns with the manuscript. I think it is currently appropriate for publication.